# Characterization of Organellar-Specific ABA Responses during Environmental Stresses in Tobacco Cells and Arabidopsis Plants

**DOI:** 10.3390/cells11132039

**Published:** 2022-06-27

**Authors:** Yuzhu Wang, Yeling Zhou, Jiansheng Liang

**Affiliations:** 1Co-Innovation Center for Modern Production Technology of Grain Crops/Jiangsu Key Laboratory of Crop Genetics and Physiology, Key Laboratory of Plant Functional Genomics of the Ministry of Education, Yangzhou University, Yangzhou 225009, China; wyz244653444@163.com; 2Key Laboratory of Molecular Design for Plant Cell Factory of Guangdong Higher Education Institutes, Department of Biology, School of Life Sciences, Southern University of Science and Technology, Shenzhen 518055, China

**Keywords:** abscisic acid, FRET-FLIM, sensor, organellar-specific, environmental stress

## Abstract

Abscisic acid (ABA) is a critical phytohormone involved in multifaceted processes in plant metabolism and growth under both stressed and nonstressed conditions. Its accumulation in various tissues and cells has long been established as a biomarker for plant stress responses. To date, a comprehensive understanding of ABA distribution and dynamics at subcellular resolution in response to environmental cues is still lacking. Here, we modified the previously developed ABA sensor ABAleon2.1_Tao3 (Tao3) and targeted it to different organelles including the endoplasmic reticulum (ER), chloroplast/plastid, and nucleus through the addition of corresponding signal peptides. Together with the cytosolic Tao3, we show distinct ABA distribution patterns in different tobacco cells with the chloroplast showing a lower level of ABA in both cell types. In a tobacco mesophyll cell, organellar ABA displayed specific alterations depending on osmotic stimulus, with ABA levels being generally enhanced under a lower and higher concentration of salt and mannitol treatment, respectively. In Arabidopsis roots, cells from both the meristem and elongation zone accumulated ABA considerably in the cytoplasm upon mannitol treatment, while the plastid and nuclear ABA was generally reduced dependent upon specific cell types. In Arabidopsis leaf tissue, subcellular ABA seemed to be less responsive when stressed, with notable increases of ER ABA in epidermal cells and a reduction of nuclear ABA in guard cells. Together, our results present a detailed characterization of stimulus-dependent cell type-specific organellar ABA responses in tobacco and Arabidopsis plants, supporting a highly coordinated regulatory network for mediating subcellular ABA homeostasis during plant adaptation processes.

## 1. Introduction

Abscisic acid (ABA) is one of the key phytohormones that is utilized by sessile plants for coping with endogenous stimuli as well as external challenges. As a multifaceted hormone, ABA accumulates in various plant tissues and controls diverse plant physiological processes, including seed maturation and stress adaptation [1,2,3,4]. In accordance with ever-changing environments, cellular ABA levels are constantly in a dynamic equilibrium balanced between synthesis, catabolism, and transport. For example, upon drought stress, ABA de novo biosynthesis is activated, starting with the epoxidation of zeaxanthin by the ABA DEFICIENT 1 (ABA1) in the plastids and ending in the cytoplasm with the oxidative cleaving of 9′-cis-neoxanthin and 9′-cis-violaxanthin by the 9-cis-epoxycarotenoid dioxygenase (NCED) being the rate-limiting step [5,6,7]. Very recently, an alternative, ABA1-independent ABA biosynthetic pathway has been unraveled in both Arabidopsis and rice [8]. In response to changes in the environment, rapid ABA accumulation can be achieved via one-step hydrolysis of ABA-GE by β-glucosidases BG1 and BG2 that occurred in the endoplasmic reticulum (ER) and vacuole, respectively [9,10]. The catabolism of ABA mediated by ABA conjugation and catalytic hydroxylation, both of which occur in the cytoplasm, also contributes to cellular ABA homeostasis substantially [11,12,13,14]. In addition, translocation of ABA among cells and tissues are essential for activation of the ABA signaling pathway in systemic stress responses [15,16]. Due to the weak-acid nature of ABA and its declined diffusion which resulted from cytosolic alkalization under stressed conditions [17], the translocation of ABA mainly relies on active transporters from the ATP-binding cassette (ABCG) family [18,19,20,21,22], NRT1/PTR (NPF) [23], multidrug and toxic compound extrusion (MATE)-type/DTX family [24], and AWPM-19 family [25].

Whilst the production of free ABA involves multiple subcellular compartments including the ER, vacuole and the cytoplasm, ABA signaling initiated upon recognition of ABA mainly takes place in the cytoplasm and the chloroplast where ABA receptors, the Pyrabactin resistance 1 (PYR1)/ PYR1-like (PYL) proteins and the Mg-chelatase H subunit (CHLH) are located, respectively [26,27]. Nuclear localization of ABA receptors has also been shown to be important for regulation of plant root growth and stress responses [28,29]. The diverse subcellular sites of ABA synthesis and its action raises an intriguing possibility that ABA in different subcellular pools may act distinctly to endo- and exogenous stimulus, resulting in specific physiological roles during plant development and stress accommodation. For example, loss of AtBG1 compromised the production of ABA in the ER as well as in the extracellular area, which was associated with reduced growth and yellow leaf phenotypes in the *bg1* mutants [10]. Further studies revealed that the BG1/BGLU18-mediated ABA production was triggered by dynamics of the leaf ER bodies under dehydration, exerting a distinct physiological impact from the de novo ABA biosynthesis [30,31]. ABA in the cytoplasm was reported to activate the glucan-hydrolyzing enzymes BAM1 and AMY3, leading to starch degradation in response to osmotic stress [32]. Cytosolic ABA levels could also be maintained through the ABA receptor RACR12/13-mediated glycosylation activity of UGT71C5 in Arabidopsis that were rehydrated after drought stress [33]. With the ABAR/CHLH presenting as a chloroplast envelope-localized ABA receptor, ABA in the chloroplast may act as a retrograde signal from plastid to nucleus during stress responses [34]. In the nucleus, ABA was found to be able to induce the accumulation of certain E3 ubiquitin ligases to either attenuate or propagate ABA signal depending upon environmental cues [35,36]. Thus, knowledge of how ABA is distributed and regulated in different organelles is essential for understanding plant adaptation via ABA signaling.

To address this, the monitoring of organellar ABA levels is of utmost importance. Cellular ABA levels had been detected through immunoassays and chromatographic methods coupled with mass spectrometry (MS) [37,38,39]. Nonetheless, the prohibitive cost as well as the irreparably destructive effect on plant materials associated with both immune-based and chromatographic approaches have limited their application for in-depth investigation of spatial and temporal regulation of ABA during plant growth and stress responses. Alternatively, ABA reporters based on the promoter of marker genes involved in ABA signaling (*pRD29A/B*, *pRAB18* and *pAtHB6*) have been developed for monitoring ABA responses in distinct tissues and cells in a real-time manner [40,41,42,43,44], although the reactions were delayed and indirect. In 2014, two fluorescence resonance energy transfer (FRET)-based ABA reporters, ABAleons [45] and ABACUSs [46], were developed for the direct monitor of endogenous ABA concentrations in different plant tissues and under stressed conditions. Both reporters were designed based on ABA-dependent conformational changes within ABA receptor complexes (PYR1/PYL1-ABA-ABI), which were reflected by alterations in the ratio of fluorescence emissions. More recently, a nanobody-epitope-based targeting strategy has been employed and the modified ABAleon2.1, named as ABAleon2.1_Tao3, was targeted to the ER membrane, allowing direct imaging of the ABA level at subcellular resolution [47]. Utilizing the ER membrane-anchored ABAleon2.1_Tao3s, the authors showed ER-specific increases in the level of ABA upon environmental stimuli [47].

In this context, we set out to characterize ABA responses in different organelles using an ABA sensor ABAleon2.1_Tao3 (Tao3) targeted to the ER, nucleus, and the chloroplast/plastid as soluble chimeric probes. The present study supports the cytoplasm as a fundamental source for ABA production and the nucleus and probably also chloroplast/plastid as unique ABA pools for distinguishing specific environmental stimulus during plant stress responses. This research provides new insights not only into ABA compartmentalization in different cells and plant species, but also into the fine-tuning of ABA at subcellular resolution during plant accommodation to its environments.

## 2. Materials and Methods

### 2.1. Plant Materials and Growth Conditions

Tobacco (*Nicotiana tabacum* L.) SR1 was grown on Murashige and Skoog (MS) medium (PhytoTech Labs, Lenexa, KS, USA) supplemented with 2% (*w*/*v*) sucrose, 0.5 g/L MES, and 0.8% (*w*/*v*) plant agar at pH 5.7 in 16-/8-h light-dark cycles at 25 °C.

The Arabidopsis genetic materials used in this study are in the Col-0 background. Seeds were sterilized in 2% sodium hypochlorite, washed three times in 70% ethanol and grown on 1/2 MS medium supplemented with 1% (*w*/*v*) sucrose, 0.5 g/L MES, and 1% (*w*/*v*) plant agar at pH 5.7 in 16-/8-h light-dark cycles at 22 °C.

### 2.2. Plasmid Preparation

All constructs used in this study were generated according to established standard procedures such as fragment clone, single or double digestion, ligation, and transformation in *Escherichia coli* DH5α. The new fragments/constructs, primers, and clone information are listed in Appendix A. Plasmids used for tobacco protoplast transfection were extracted using plasmid DNA Maxi kits (OMEGA Bio-tek, Norcross, GA, USA).

### 2.3. Protoplast Isolation and Transfection

Isolation of tobacco mesophyll protoplasts were performed as previously described [47,48]. Briefly, protoplasts were prepared from perforated tobacco leaves by overnight incubation in the buffer (3.05 g/L Gamborg B5 salt medium (PhytoTech Labs, Lenexa, KS, USA), 500 mg/L MES, 750 mg/L CaCl_2_·2H_2_O, 250 mg/L NH_4_NO_3_ adjusted to pH 5.7 with KOH, supplemented with 0.2% *w*/*v* macerozyme R-10 and 0.4% *w*/*v* cellulose R-10) at 25 °C in the dark. They were rebuffered through washing three times in a 40 mL electrotransfection buffer (137 g/L sucrose, 2.4 g/L HEPES, 6 g/L KCl, 600 mg/L CaCl_2_·2H_2_O adjusted to pH 7.2 with KOH). Five hundred microliters of protoplasts in a total volume of 600 μL electrotransfection buffer were electrotransfected with 2-6 μg plasmid DNA using the Gene Pulser Xcell^TM^ (Bio-Rad Laboratories, Hercules, CA, USA) with a pulse at 160 V for 10 ms. After transfection, each sample was supplemented with a 2 mL incubation buffer and incubated for 16–24 h at 25 °C in the dark.

### 2.4. Confocal Laser Scanning Microscopic Analysis

Tobacco protoplast imaging was performed using a Nikon A1plus confocal laser scanning microscope (Nikon, Melville, NY, USA) with a 40× (1.15 numerical aperture) water immersion objective. Tobacco epidermal cells, Arabidopsis leaves and roots imaging was performed using a Leica TCS SP8 confocal laser scanning microscope (Leica Microsystems, Durham, NC, USA), with a 20× air objective. The sensor proteins and RFP markers were excited with 488 nm and 561 nm, and emission at the range of 500–550 nm and 570–620 nm was detected, respectively. Pinholes were adjusted to 1 Airy unit for each wavelength. Post-acquisition image processing was performed with ImageJ.

### 2.5. Fluorescence Lifetime Imaging Microscopy

FLIM recordings were performed at a Nikon A1R equipped with a PicoHarp time-correlated single-photon counting (TCSPC) module and a PDL800-D multichannel picosecond pulsed diode laser driver (PicoQuant, Berlin, Germany). The donor τ_mT_ was excited with a 440 nm laser at a 20 MHz pulse frequency. Emissions was recorded at 482/35 nm by TCSPC until reaching a count of at least 250 photons in the brightest pixel (total count of at least 10,000 photons). At least 10 cells per sample per treatment were recorded. For time-resolved single cell recordings, cells were mixed gently and mounted to the slides immediately after treatment. FLIM recording started at 5 min after treatment and at 5 min intervals. For time-resolved root single cell recordings, seedlings were immersed in treatment buffer and mounted to the slides immediately. FLIM recording started at 10 min after treatment and continued at 10-min intervals.

FLIM data were analyzed using SymphoTime64 v2.0 (PicoQuant, Berlin, Germany). Values were collected based on the specifically selected fluorescence signals in each cell using the software’s “region of interest” (ROI) tools to avoid background noise. To calculate fluorescence lifetimes of the donor τ_mT_, TCSPC histograms were reconvoluted with an instrumental response function and fitted against a mono- (in the absence of the acceptor) or multiexponential (in the presence of the acceptor, *n* = 2) decay model. Only fittings giving Chi squared values between 0.9 and 1.2 were considered.

The FRET efficiency (E) was obtained either by the formula E = (1 − τ_DA_/τ_D_) × 100%, in our case, E = (1 − τ_mT-Tao3_/τ_mT-mT_) × 100%, where τ_mT-Tao3_ means the fluorescence lifetime of mTurquoise in Tao3s and τ_mT-mT_, the fluorescence lifetime of mTurquoise in donor-only fusion proteins, or directly from the lifetime-based FRET fitting module in SymphoTime64. During the FRET fitting, the τ_D_ is set based on the calculated τ_mT_ and the multiexponential (*n* = 2) decay model is applied.

### 2.6. Stress Assays Using Tobacco Protoplasts

Protoplasts that had been electrotransfected with the same plasmids were collected as described above. The floating cells were transferred to a new tube to remove cell debris and washed twice.

For the exogenous chemical assays, protoplasts were aliquoted into 0.2 mL per tube. Stock solutions of mannitol (800 mM), and NaCl (200 mM) were prepared in an incubation buffer. For osmotic stress assays, 0.1 mL incubation buffer containing certain amounts of salt or mannitol was added to each protoplast aliquot. For stress recovery assays, the mannitol-or salt-containing buffer were aspirated, and the protoplasts were resuspended with 0.3 mL incubation buffer for 2 h.

### 2.7. Plant Transformation and Stress Treatments

Organellar Tao3s were transformed into Arabidopsis Col-0 by the floral dip method [49]. Transformants were selected on 1/2 MS medium supplemented with 50 μg/mL kanamycin. Positive lines were further selected by fluorescence signals under the Nikon A1plus confocal microscope.

For mannitol treatments, T2 seeds were sown on 1/2 MS medium after sterilization with 1.5% (*v*/*v*) sodium hypochlorite. Five-day-old seedlings were transferred to a 12-well plate containing 1.6 mL 1/2 MS liquid medium plus kanamycin in each well for 48 h. About 0.4 mL water containing mannitol was added to each well, making a final concentration of mannitol at 50 mM, 100 mM, and 200 mM, respectively. About 0.4 mL water was added in parallel as mock treatment. For stress recovery, the mannitol-treated seedlings were transferred to the mock well for 2 h. Seedlings were subjected to FLIM recording 1 h later. At least three seedlings were recorded in each line for each treatment, and at least three independent lines per transformation were used.

### 2.8. Phenotypic Analysis of Transgenic Plants

For analysis of ABA sensitivity, five-day-old seedlings grown in 1/2 MS plates were transferred to media containing 0 or 10 μM ABA. For the mannitol tolerance assay, five-day-old seedlings were transferred to 1/2 MS medium containing 0- or 300-mM mannitol. The roots were imaged on the day before and five days after transfer using a Canon EOS 800D. The root length was analyzed by ImageJ. The relative root length was calculated as the elongated root length at day five relative to the root length at day 0. The experiment was performed three times and each replicate included eight plants per genotype.

## 3. Results

### 3.1. Targeting of ABAleon2.1_Tao3 to Different Suborganelles as Soluble Proteins

We previously reported the targeting of ABAleon2.1_Tao3 (Tao3) to the endoplasmic reticulum (ER) membrane through a nanobody-epitope-mediated interaction [47]. To target the ABA sensor to other suborganelles like chloroplasts and the nucleus, we employed the traditional targeting strategy by the addition of organellar-specific signal peptide (Figure 1A). For the ER luminal targeting, an N-terminal signal peptide and C-terminal H/KDEL sequence was fused to Tao3 [50], whereas the nuclear targeting involves the fusing of Tao3 with a simian virus 40 (SV40) signal [51] and a dual BAM4-derived sequence for chloroplast targeting [52].

To validate correct organellar targeting of Tao3s, we co-expressed the organellar Tao3s with their corresponding membrane RFP markers in both tobacco epidermal and mesophyll cells. A confocal microscopic analysis consistently showed the overlapped fluorescent signals from Tao3s and their respective RFP in both cell types (Figure 1B), confirming the presence of Tao3 in the four suborganelles.

### 3.2. Distinct Subcellular ABA Distribution Pattern in Different Tobacco Cells

As shown before, Tao3 responded to exogenous ABA treatment with specific increases in the fluorescence lifetime of the donor fluorophore mTurquoise (τ_mT_) [47]. To test if the other three organelle-targeted soluble Tao3s respond similarly to ABA, we generated their corresponding donor-only counterparts using the same targeting sequences as those applied for Tao3s (Figure 2A). Indeed, like cytosolic Tao3, the ER-, nuclear-, and chloroplast targeted Tao3 exhibited notable increases in the τ_mT_, whereas their donor-only counterparts displayed little changes in response to ABA application (Appendix A), supporting the substrate responsiveness of all four organellar targeted Tao3s.

For FRET sensors, substrate binding often resulted in alterations in FRET efficiency (E) that can be measured through ratiometric emission, photobleaching FRET, and fluorescence lifetime imaging microscopy (FLIM). In the case of FLIM-FRET, E is described by the differences of the fluorescence lifetime of the donor in the presence (τ_DA_) and absence of the acceptor (τ_D_) [53]. FLIM analysis revealed that both tobacco epidermal and mesophyll cells showed similar levels of τ_D_ in the cytosol, ER lumen and nucleus, whereas the τ_D_ in the chloroplast was strikingly lower than that in the other three organelles (Figure 2B,C). As the τ is inherent in a fluorescent protein and sensitive to its surrounding [54], the notably reduced τ_D_ in the chloroplast could be attributed by the differed pH (around 8 in chloroplast stroma vs. 7.3 in cytoplasm) [55,56] and much higher viscosity in the chloroplasts [57].

The presence of ABA caused changes in the conformation of ABAleon2.1 and Tao3, leading to increases in the distance or dipole orientations between the coupled mTurquiose and cpVenus and ultimately resulting in reductions in the E [45,47]. Thus, for FRET-based ABA sensor ABAleon2.1 and Tao3, the endogenous ABA level is inversely related to the E value. Our lifetime-based FRET fittings on the Tao3s targeted to the four organelles in tobacco epidermal cells revealed notably reduced E in the ER and the nucleus, whereas the E in the chloroplast was significantly higher compared to that in the cytoplasm (Figure 2D). This indicated a higher level of ABA in the ER and the nucleus and a lower level of ABA in the chloroplast compared to that in the cytosol in tobacco epidermal cells. Interestingly, in tobacco mesophyll cells, while the E was at a similar level in both the cytosol and the ER, it was greatly enhanced in the nucleus and the chloroplast (Figure 2E), implying a much lower level of ABA in those two organelles in tobacco mesophyll cells. Besides, it was noticed that organellar ABA in mesophyll cells displayed generally lower E ranging between 5–10% compared with those in epidermal cells with the E ranging between 10–20% (Figure 2D,E), indicating a general higher level of ABA in the mesophyll cells. This is plausible given that mesophyll cells are the major sites for ABA biosynthesis in water-stressed leaves [58]. Together, the four organellar Tao3s show the distinct distribution of ABA at subcellular scale in different tobacco cells.

### 3.3. Organellar ABA Displays Distinct Response Pattern upon Salt Stress in Tobacco Protoplasts

We have previously shown ER-specific responses in the level of ABA upon environmental stresses in tobacco protoplasts [47]. To investigate how ABA in other organelles responds to different environmental stimuli, we first applied a series of concentration of NaCl to organellar Tao3s-transfected cells to mimic different amounts of salt stress. Consistent with the ER membrane-anchored Tao3s showing increases of ABA in the cytoplasm but not in the ER upon 10 mM NaCl treatment [47], the soluble Tao3s hinted at a similar response pattern of cytosolic and ER ABA (Figure 3A,B). While their donor-only counterparts did not respond to the salt stress (Appendix A), the increases in the τ_mT_ of the soluble Tao3s corresponded to elevations in the level of ABA. Intriguingly, when applied to higher concentrations of NaCl (50 mM and 100 mM), tobacco protoplasts exhibited little changes in cytosolic ABA compared to those non-stressed cells (Figure 3A,B). Moreover, ABA in the nuclear and chloroplast displayed similar responses as that in cytoplasm, albeit with the level of ABA still being considerably enhanced in the chloroplast upon higher salt treatment (Figure 3A,B). The extensive enhancement in the level of organellar ABA triggered exclusively upon a lower concentration of NaCl indicates that ABA plays an important role during mild but not severe salt stress responses.

### 3.4. Organellar ABA Shows Divergent Responses upon Osmotic Stress in Tobacco Protoplasts

To understand how organellar ABA responds to different levels of osmotic stress, we applied series concentrations of mannitol to the organellar Tao3 transfected cells. Again, the donor-only counterparts showed little responses in their τ_mT_ to the mannitol treatment (Appendix A), whereas the soluble Tao3s responded to mannitol in a concentration-dependent manner (Figure 4A,B), implying that the responses are specific to the mannitol treatment. More specifically, cytosolic, ER and chloroplast ABA were steadily enhanced when applied to increasing concentrations of mannitol, albeit with ER ABA being significantly elevated at a concentration of 100 mM and chloroplast ABA being notably enhanced at 200 mM mannitol treatment (Figure 4A,B). In contrast, the nuclear ^NLS^Tao3 responded to increasing mannitol with decreasing τ_mT_ (Figure 4A,B), implying a steady reduction in the level of ABA in the nucleus triggered upon increasing the level of osmotic stresses.

During plant responses to environmental stresses, the timely and efficient recovery involving the drop of cellular ABA levels back to the normal level is critical for stress adaptation [59]. To understand if it is true that the stress-induced ABA returns to normal in the suborganelles, we performed washout assays for imitation of stress relief. Indeed, both mannitol-induced ABA in the cytosol and in the ER were back to basal levels when the osmotic stimulus was washout (Figure 4C,D). Surprisingly, the reduction of ABA in the nucleus triggered upon mannitol treatment was reversed with an enhancement of nuclear ABA when the stress was relieved (Figure 4E). In the chloroplast, the relatively mild osmotic stress triggered a slight increase in the level of ABA, which also shrank upon stress recovery (Figure 4F).

### 3.5. Time-Resolved Analysis of Organellar Tao3s Shows Inversely Altered ABA in the Cytosplasm and the Nucleus during Rapid Osmotic Stress Responses

Previous time-resolved analysis of ER membrane-targeted Tao3s showed distinct responses of ABA in the cytoplasm and the ER lumen within one hour upon environmental stimuli [47]. It was noticed that the most prominent responses of ABA occurred within 30 min (min) upon stress. We thus recorded the rapid responses of ABA in the four organelles 30 min after mannitol treatment. Indeed, within 30 min upon mannitol application, the τ_mT_ was evidently enhanced compared to the mock treatment (Figure 5A), implying the rapid enhancement of ABA in the cytoplasm triggered upon osmotic stress. Whilst the τ_mT_ in the ER was fluctuated in a less pronounced way (Figure 5B), that in the nucleus was strikingly reduced at 30 min after treatment (Figure 5C), indicating an osmotic stress-induced swift reduction of nuclear ABA. Similar to its cytosolic counterpart, the τ_mT_ in the chloroplast gradually increased by the treatment of mannitol (Figure 5D), suggesting the enhancement of ABA level in the chloroplast induced by osmotic stress.

### 3.6. Targeting of Tao3s to Different Organelles in Arabidopsis Plants Reveals Osmotic Stress-Induced Exclusive Accumulation of ABA in the Cytosol in Root Meristem Zone Cells

As Tao3 manifested ABA responsiveness with evident increases in the τ_mT_ when transformed into Arabidopsis plants [47], we generated the other three organellar, including ER, nuclear and plastid/chloroplast Tao3-expressing plants for investigation of ABA compartmentalization under stress conditions. Confocal microscopic analysis showed clear fluorescent signals from the four organellar Tao3s in both the root and leaf tissues, corresponding to the localization of the probes in their respective compartments (Appendix A).

As one of the primary stress sites, plant root acts importantly for sensing and adaptation to changes in the environment [60]. To gain a more detailed understanding of organellar ABA responses under stressed conditions, we analyzed organellar Tao3s in different root cells. The series application of mannitol to the organellar Tao3s-expressing seedlings resulted in notable increases in the τ_mT_ of cytosolic Tao3 but not in that of other three organellar Tao3s (Figure 6A,B), indicating an exclusive accumulation ABA in the cytosol in root cells from the meristem zone, wherein the ER and nuclear ABA remained unaltered. In the plastid, the τ_mT_ was not changed with remarkable reduction until the exogenous osmolyte reached a highest level (200 mM) (Figure 6A,B), implying the efflux or consumption of ABA in the plastid upon severe osmotic stresses. Further stress washout assays support the exclusive accumulation of ABA in the cytosol triggered upon osmotic stress (Figure 6C–F) and that returned to a basal level when the osmolyte was washed away (Figure 6C).

### 3.7. Organellar Tao3s Show Organellar Specific Changes in the Level of ABA Induced upon Osmotic Stress in Arabidopsis Root Elongation Zone

Next, we examined the organellar Tao3s in the root elongation zone under the same treatments. It was noticed that mannitol treatment substantially enhanced the τ_mT_ of cytosolic Tao3 at a relatively lower level (50 mM mannitol), whereas it did not influence the ER-targeted Tao3 at all levels (Figure 7A,B). In contrast, the nuclear- and plastid-resided Tao3 displayed notable reductions in the τ_mT_ upon mannitol treatment, albeit the τ_mT_ of the chloroplast Tao3 remained at the same level at the higher level of mannitol treatment as those unstressed (Figure 7A,B). These results show an elevating and decreasing response pattern of cytosolic and nuclear/plastid ABA upon osmotic stress in root cells from the elongation zone, respectively. Stress recovery assays supported the stress-triggered respective mode of ABA in each organelle, all of which returned to the basal level when the cells were recovered from stress (Figure 7C–F).

### 3.8. Organellar Tao3s Reveal Subtle Changes in Arabidopsis Leaf Cells upon Osmotic Stress

ABA accumulates in both roots and shoots upon water deficit stress [7,15,58,61]. To see how organellar ABA behaves in leaf tissue when exposed to stress, we treated the five-day-old seedlings with 100 mM mannitol as mild osmotic stress. FLIM analysis showed that in Arabidopsis epidermal cells, mannitol treatment caused a notable increase of the τ_mT_ in the ER but not in the other three organelles (Figure 8A), indicating a rise of ER ABA level triggered upon osmotic stress. In guard cells, osmotic stimulus did not increase the τ_mT_ in all the four organelles, but rather, it decreased considerably the τ_mT_ in the nucleus (Figure 8B), implying a reduction of ABA molecules in the nucleus. The overall hyposensitive responses of organellar ABA presented here could be associated with the relatively lower ABA affinity of Tao3 [47].

### 3.9. Cytosolic and ER Tao3-Expressing Arabidopsis Plants Exhibit Altered Sensitivity to ABA and Mannitol

Introduction of FRET-based ABA probes, such as ABAleon2.1 and ABACUS, into Arabidopsis plants was reported to affect the overall ABA sensitivity [45,46]. To examine whether the effect on ABA responses by the expression of ABA sensor persists when targeted to organelles, we analyzed the root growth of four organellar Tao3-expressing plants in comparison with that of the Col-0 wild-type. As the expression level of sensors would affect the responsiveness [45], we chose the seedlings with similar fluorescence emission for downstream application. It was noticed that expression of cytosolic and ER Tao3 in Arabidopsis plants had little effect on primary root growth, whilst the nuclear and chloroplast/plastid Tao3 plants exhibited notably reduced growth (Figure 9A,D). ABA-inhibited root growth was observed in all investigated lines, with the cytosolic and ER Tao3 plants being less sensitive compared to other lines (Figure 9B,E). This is consistent to the ABA hyposensitivity found in ABAleon2.1 plants [45]. Intriguingly, when exposed to 0.3 M mannitol treatment, Tao3 plants displayed an enhanced sensitivity with respect to primary root growth compared to Col-0 and other organellar Tao3 plants (Figure 9C,E).

## 4. Discussion

Phytohormones like ABA act in integrative networks with multiple sites of synthesis and action. For many years, the regulation of ABA homeostasis and signaling during stress responses has been extensively studied in various tissues/cells, with the guard cells attracting the most attention [61,62,63,64]. Knowledge in stress-triggered ABA responses at subcellular scale remain scarce owing to the lack of effective sensors for monitoring instantaneous ABA concentration. Here we show the targeting of the previously constructed ABA sensor Tao3 [45,47] to different subcellular compartments and application of the organellar-resided Tao3s reveals compartment-specific ABA responses triggered upon distinct environmental cues in tobacco and Arabidopsis plants.

As a weak acid, cellular ABA occurs either as protonated forms or dissociated anion forms with the capacity to diffuse freely across membranes or not. Owing to this, it was proposed that subcellular distribution of ABA is controlled by the pH in different compartments [65]. The mathematical model based on experimental data from *Valerlanella locusta* had predicted that the chloroplasts contained the highest ABA content irrespective of the cell type [66]. In contrast, our organellar targeted Tao3s pointed to a lowest ABA level in the chloroplasts compared to the other three compartments in both the tobacco epidermal and mesophyll cells (Figure 2D,E). As we employed a BAM-derived sequence for chloroplast targeting, it is likely that that the final location of the ^cTP^Tao3 is somewhere between the inner and outer membrane of the chloroplast. Thus, instead of the ABA level in the chloroplast stroma, the ^cTP^Tao3 estimates ABA distribution around the chloroplast membrane. Besides, organellar Tao3s revealed a relatively higher ABA concentration in the ER in both tobacco cells, albeit the differences between the cytoplasm and the ER in the mesophyll cells were less prominent (Figure 2D,E). This is consistent with our previous observations showing by the ER membrane anchored Tao3s and the insignificance of ABA level between the ER and the cytosol is likely attributed by a reduced affinity for ABA in soluble Tao3s [47]. Intriguingly, while the nuclear ABA was essentially at the similar level (with the average E around 10%) in both cell types, a lower E level associated with a higher level of ABA in the cytosol was obtained in mesophyll cells (Figure 2D,E), rendering a contrasting pattern of nuclear ABA in the two different cell types. Considering that epidermal cells act as the first barrier when exposed to an adverse environment, it is logical for them to hold a relatively higher ABA level in the nucleus so that when stress came, sufficient ABA signaling could be rapidly initiated for downstream transcriptomic changes and ultimately for stress adaptation.

As a stress hormone, cellular ABA has been shown to accumulate in response to adverse environments [67]. Stress assays with our organellar Tao3s supported an organellar-specific enhancement of ABA depending upon stress stimulus. For example, a general enhancement of ABA concentration occurred extensively in all the four organelles in response to a lower concentration of salt treatment while osmotic stress induced a dose-dependent increase of ABA in all organelles except in the nucleus (Figure 3 and Figure 4). Given the fact that salinity usually induces multiple cellular stresses including osmotic stress, ionic stress and oxidative stress [68], higher salinity conditions in association with overwhelming ionic and oxidative stress trigger either a higher rate of ABA catabolism than of synthesis [13,14,69], or consumption of massive ABA for the activation of ABA signaling, resulting in the undetectable changes in the ABA level in cytosol and in the nucleus (Figure 3B). Since our data was derived from a single timepoint measurement, it cannot be excluded that the higher salinity-induced ABA may peak at a different timepoint with enhanced magnitude. Also, an ABA-independent signaling might dominate in tobacco protoplasts exposed to higher salinity [70].

Given the cell-type specific distribution of nuclear ABA in different tobacco cells (Figure 2D,E), the inverse response pattern featured by nuclear ABA with enhancement and steady decreases upon salt (Figure 3) and osmotic stress in different cells/tissues (Figure 4, Figure 5, Figure 7B and Figure 8B), respectively, denotes a novel role of nuclear ABA for distinguishing specific cell types and stimulus during cell differentiation and stress responses [71]. In comparison, it was noticed that ABA in the cytoplasm was extensively enhanced upon different stimuli in both tobacco and Arabidopsis cells, highlighting the cytoplasm as an indispensable ABA source during stress responses.

In Arabidopsis plants, root cells from the elongation/maturation zone responded more actively to osmotic stimulus compared with the cells from the meristematic zone, with nuclear and chloroplast ABA being notably reduced upon stimulus (Figure 6 and Figure 7). This is sensible considering that whilst the root meristem is mostly associated with the brassinosteroid pathway [72], ABA signaling coupled with auxin responsible for regulating root growth mainly occurred in the root elongation/maturation zone [73,74]. Indeed, it has been shown that the water potential gradient elicited ABA-dependent differential growth response mainly in the transition and elongation zones [74]. Given the occurrence of ABA receptors in the nucleus and the chloroplast [27,75,76], the notable reductions of the nuclear and the plastid ABA indicate an active ABA signaling in this area of the roots.

In Arabidopsis leaves, organellar ABA seemed to respond reluctantly to osmotic stimulus in both the epidermal and guard cells (Figure 8A,B). While this could be partly attributed to the relatively lower ABA affinity of Tao3s, it has also been shown that salt stress induced ABA accumulation to a greater extent in roots than in shoots [4]. Despite that, osmotic stress induced a notable enhancement of ER ABA in the epidermal cells in parallel with a reduction of nuclear ABA in guard cells (Figure 8A,B), suggesting ABA production in the ER and an intensive activation of nuclear ABA signaling in the corresponding cells. Thus, it is likely that, as the central regulator of guard cell physiology [77], ABA is rapidly synthesized in the ER of epidermal cells through one-step hydrolysis upon reception of stresses [10], which is transported to the neighboring guard cells. When they enter the nucleus, a large quantity of ABA is consumed and activate a downstream signaling cascade, ultimately leading to stomatal closure [62].

The analysis of organellar Tao3-expressing Arabidopsis plants disclose a severe growth defect of ^NLS^Tao3 and ^cTP^Tao3 plants (Figure 9A,D). As Tao3s are expected to sequester a certain amount of ABA like ABAleon2.1 [45], the growth arrest by ^NLS^Tao3 and ^cTP^Tao3 implies the importance for maintaining a certain level of ABA in the nucleus and the chloroplast during plant development. Considering the role of basal ABA for supporting plant growth and development [78], it is reasonable to reckon that this beneficial effect is mainly achieved by ABA in those two compartments. Moreover, like ABAleon2.1 plants [45], cytosolic and ER Tao3 plants showed ABA hyposensitivity regarding the root growth (Figure 9B,E). The sequestration of cytosolic ABA that occurred in cytosolic Tao3 plants likely caused a reduction of free ABA in the cytosol, thus mimicking an ABA-deficient mutant. Indeed, like the *los5*/*aba3* mutant showing compromised tolerance to salt and drought stress [79], cytosolic Tao3 plants presented a hypersensitivity to osmotic stress with enhanced root growth reduction compared to wild-type Col-0 plants (Figure 9C,E). These together demonstrate that, although nuclear and chloroplast ABA are more closely related to endo-stimulus during plant development, ABA in the cytoplasm and ER are more tightly controlled for activating ABA signaling in response to exo-stimulus.

## 5. Conclusions

Collectively, we successfully targeted the previously developed ABA sensor ABATao3 to the four organelles, allowing a comprehensive characterization of the subcellular ABA distribution and responses *in planta*. The cell-type specific distribution and variations in organellar ABA upon distinct environmental cues substantiate the existence of a complicated regulatory network for tailoring compartment-specific responses in plants. Further studies with organellar Tao3s in different genetic backgrounds would aid in the identification of novel components in stress-mediated ABA signaling or transporters at the subcellular endomembrane.

## Figures and Tables

**Figure 1 cells-11-02039-f001:**
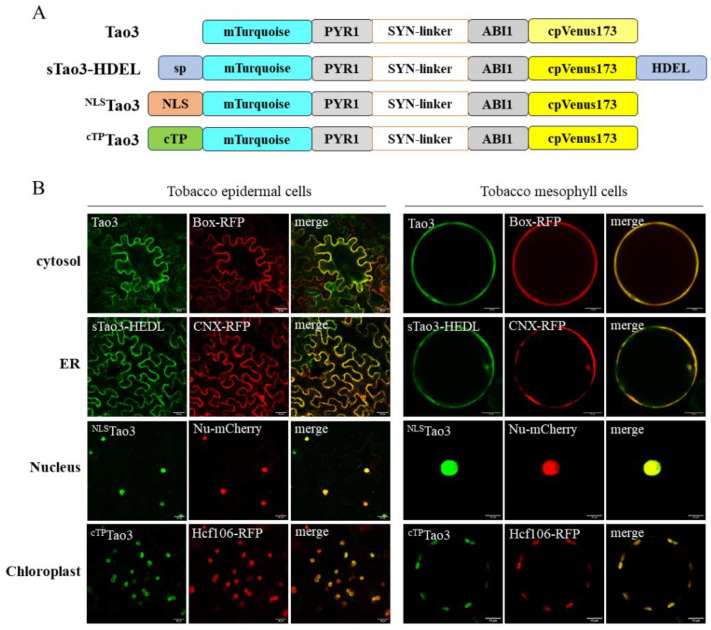
Targeting of ABA sensor ABAleon2.1_Tao3s (Tao3s) to different subcellular organelles. (**A**) Schematic representation of cytosolic (Tao3), ER (sTao3-HDEL), nucleus (^NLS^Tao3), and chloroplast (^cTP^Tao3) ABA sensor Tao3; (**B**) Co-localization of organellar-targeted Tao3s with their corresponding markers in tobacco epidermal cells and mesophyll protoplasts. Box-RFP, cytosolic RFP marker; CNX-RFP, ER membrane RFP marker; Nu-mCherry, nucleus marker; Hcf106-RFP, chloroplast marker. Scale bar, 30 μm (left), 10 μm (right).

**Figure 2 cells-11-02039-f002:**
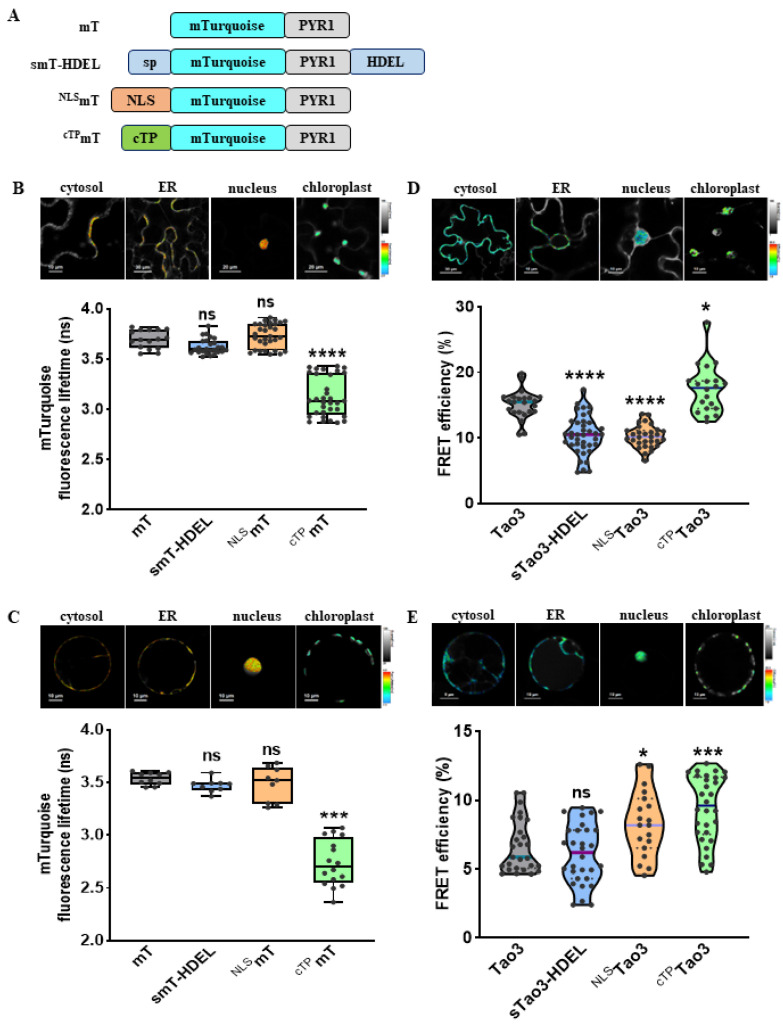
Organellar-targeted ABA sensors reveal distinct subcellular ABA distribution patterns in different tobacco cells. (**A**) Schematic representation of donor constructs targeted to the cytosol (mT), ER (smT-HDEL), nucleus (^NLS^mT), and chloroplast (^cTP^mT); (**B**,**C**) Representative pseudo-color images and data recorded by FLIM showing the fluorescence lifetime of the donor mT (τ_mT_) targeted to the four organelles in tobacco epidermal cells (**B**) and tobacco mesophyll protoplasts (**C**). For better visualization and comparisons, the color-scale bar of all the FLIM images were normalized to 2.5–4.0 and the intensity bars were from 1 to 100–200 depending on the expression level of proteins in the cells. FLIM data are presented as box plots showing all data points. Significance was calculated using one-way ANOVA followed by Dunnett’s multiple comparisons test (*** *p* < 0.001; **** *p* < 0.0001; ns, not significant). (**D**,**E**) Representative FRET images and data showing organellar ABA levels in tobacco epidermal cells (**D**) and tobacco protoplasts (**E**). FRET efficiency values are indicated as violin plots with all data points. Significance was calculated using one-way ANOVA followed by Dunnett’s multiple comparisons test (* *p* ≤ 0.05; *** *p* < 0.001; **** *p* < 0.0001; ns, not significant); Significance was calculated using Student’s *t* test (* *p* < 0.05; ns, not significant).

**Figure 3 cells-11-02039-f003:**
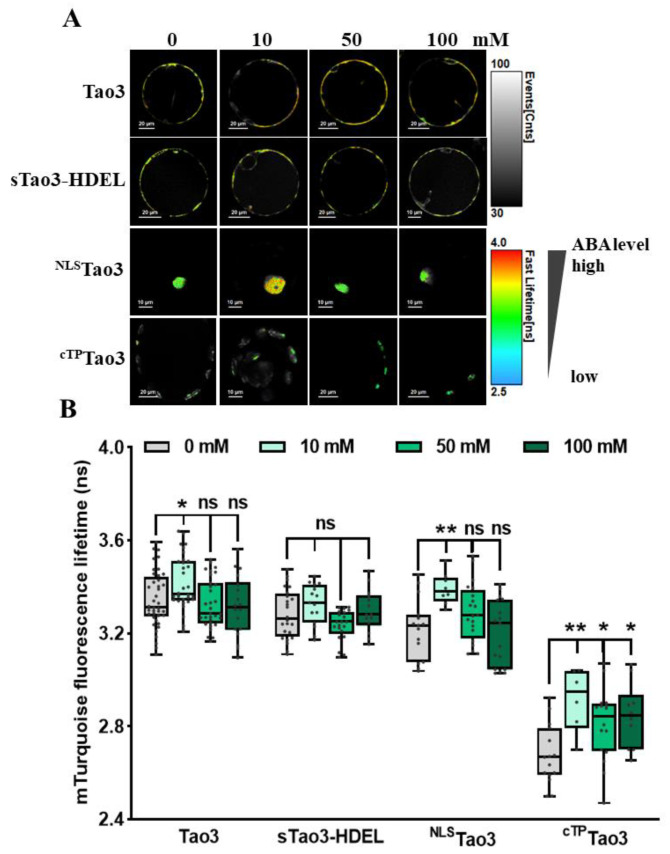
Organellar Tao3s reveal distinct changes in the ABA content in different organelles upon salt treatment in tobacco protoplasts. (**A**,**B**) Representative FLIM images (**A**) and data (**B**) showing the changes of ABA content in the four organelles at 6 h upon series concentration of NaCl treatment (0, 10 mM, 50 mM, 100 mM). FLIM data are presented as box plots showing all data points. Significance was calculated using one-way ANOVA followed by a Dunnett’s multiple comparisons test (compared within the same organelle; * *p* < 0.05; ** *p* < 0.01; ns, not significant).

**Figure 4 cells-11-02039-f004:**
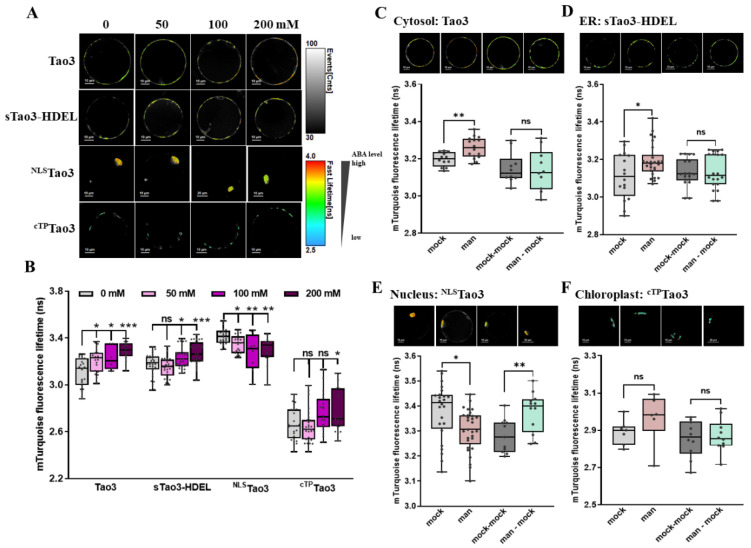
FLIM-FRET analysis reveals distinct changes in the ABA content in different organelles upon mannitol treatment in tobacco protoplasts. (**A**,**B**) Representative FLIM images (**A**) and data (**B**) showing the changes of ABA content in the four organelles at 6 h upon series concentration of mannitol (man) (0, 50, 100, 200 mM); (**C**–**F**) Representative FLIM images and data showing specific responses of the τ_mT_ in cytosolic Tao3 (**C**), ER sTao3-HDEL (**D**), nucleus ^NLS^Tao3 (**E**), chloroplast ^cTP^Tao3 (**F**) at 4 h after 100 mM mannitol treatment and washout for 2 h. FLIM data are presented as box plots showing all data points. Significance was calculated using one-way ANOVA followed by Dunnett’s multiple comparisons test (**B**) and Student’s *t* test (**C**–**F**) (* *p* < 0.05; ** *p* < 0.01; *** *p* < 0.001; ns, not significant).

**Figure 5 cells-11-02039-f005:**
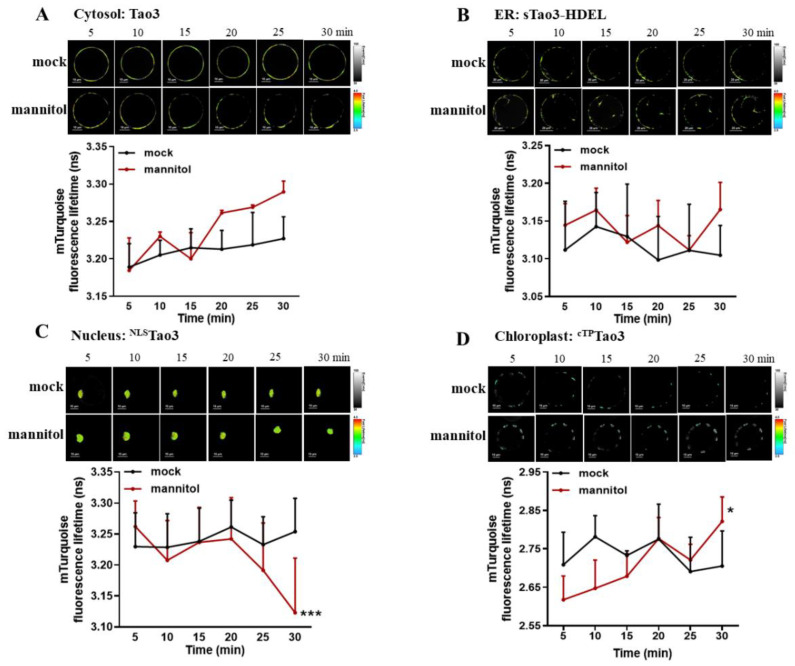
Time-resolved analysis of orgnellar Tao3s in response to osmotic stresses in tobacco protoplasts. (**A**–**D**) Successive FLIM images (up) and time-resolved recording (down) of cytosolic Tao3 (**A**), sTao3-HDEL (**B**), ^NLS^Tao3 (**C**) and ^cTP^Tao3 (**D**) within 30 min upon treatment with 200 mM mannitol and mock treatment. FLIM values were recorded from 5 min until 30 min after treatment and at 5 min intervals. FLIM data are presented as means ± SE. *n* = 6–8 cells. Significance was calculated using two-way ANOVA followed by Dunnett’s multiple comparisons test compared within the same treatment (* *p* < 0.05; *** *p* < 0.001).

**Figure 6 cells-11-02039-f006:**
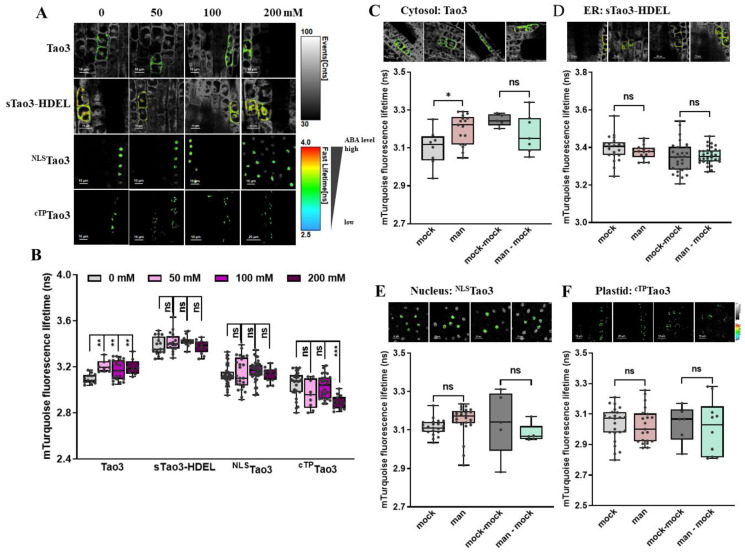
FLIM-FRET analysis reveals osmotic stress triggered-distinct changes in the ABA content in different organelles in Arabidopsis root meristem cells. (**A**,**B**) Representative FLIM images (**A**) and data (**B**) showing the changes of ABA content in the four organelles at 6 h upon series concentration of mannitol (man) (0, 50, 100, 200 mM); (**C**–**F**) Representative FLIM images and data showing specific responses of the τ_mT_ in cytosolic Tao3 (**C**), ER sTao3-HDEL (**D**), nucleus ^NLS^Tao3 (**E**), plastid ^cTP^Tao3 (**F**) at 4 h after 100 mM mannitol treatment and washout for 2 h. FLIM data are presented as box plots showing all data points. Significance was calculated using one-way ANOVA followed by Dunnett’s multiple comparisons test (**B**) and Student’s *t*-test (**C**–**F**) (* *p* < 0.05; ** *p* < 0.01; *** *p* < 0.001; ns, not significant).

**Figure 7 cells-11-02039-f007:**
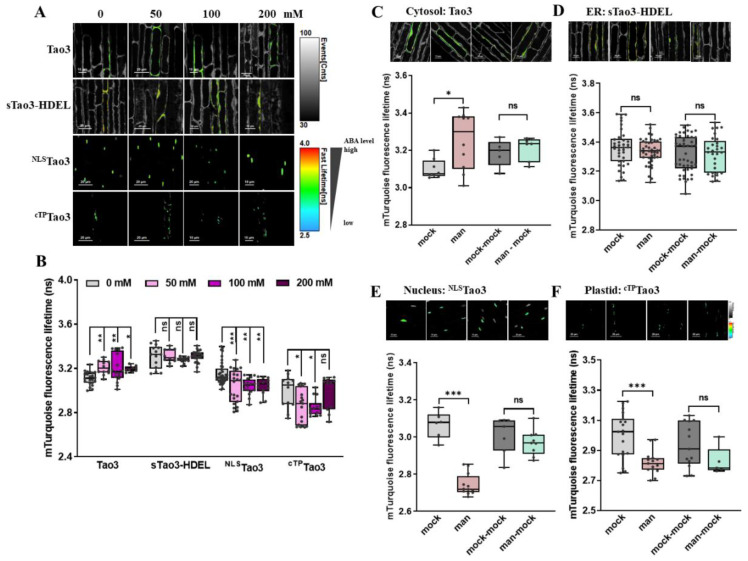
FLIM-FRET analysis reveals distinct alterations in organellar ABA upon mannitol treatment in cells from the Arabidopsis root elongation zone. (**A**,**B**) Representative FLIM images (**A**) and data (**B**) showing the changes of ABA content in the four organelles at 6 h upon series concentration of mannitol (man) (0, 50, 100, 200 mM); (**C**–**F**) Representative FLIM images and data showing specific responses of the τ_mT_ in cytosolic Tao3 (**C**), ER sTao3-HDEL (**D**), nucleus ^NLS^Tao3 (**E**), Plastid ^cTP^Tao3 (**F**) at 4 h after 100 mM mannitol treatment and washout for 2 h. FLIM data are presented as box plots showing all data points. Significance was calculated using one-way ANOVA followed by Dunnett’s multiple comparisons test (**B**) and Student’s *t*-test (**C**–**F**) (* *p* < 0.05; ** *p* < 0.01; *** *p* < 0.001; ns, not significant).

**Figure 8 cells-11-02039-f008:**
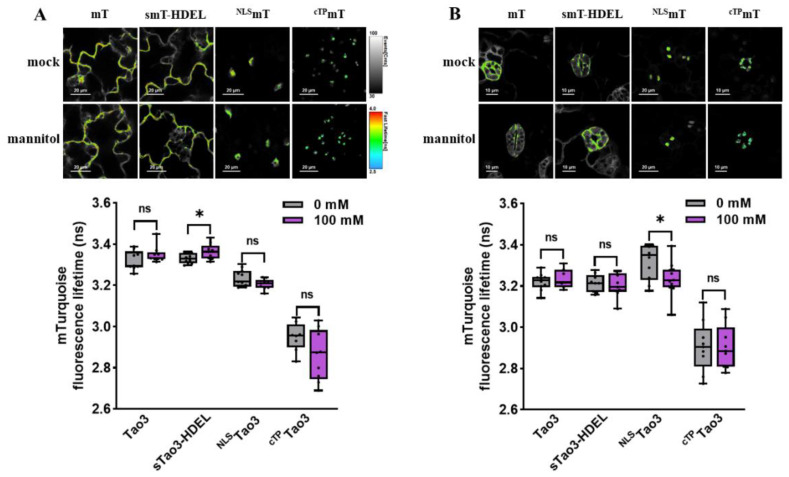
FLIM-FRET analysis reveals organellar-specific changes in the level of ABA in Arabidopsis leaf cells upon mannitol treatment. (**A**,**B**) Representative FLIM images and data showing the changes of ABA content in four organelles at 6 h upon 100 mM mannitol treatment in epidermal cells (**A**) and guard cells (**B**). FLIM data are presented as box plots with all data points. Significance was calculated using Student’s *t* test (* *p* < 0.05; ns, not significant).

**Figure 9 cells-11-02039-f009:**
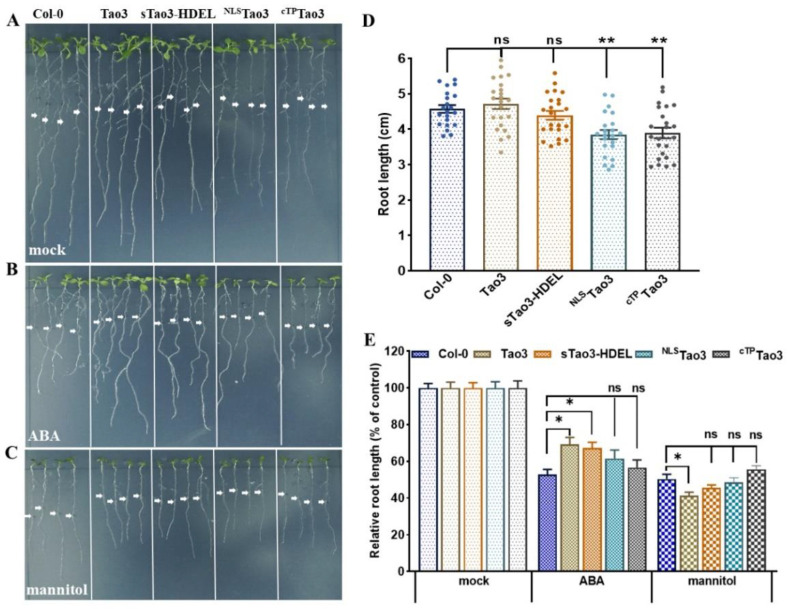
Cytosolic and ER Tao3-expressing plants show a hypo- and hypersensitivity to ABA and mannitol, respectively. From left to right, Col-0 wild type, Tao3, sTao3-HDEL, ^NLS^Tao3 and ^cTP^Tao3. (**A**–**C**) Representative images showing 10-day-old seedlings five days after transfer to ½ MS medium supplemented with mock (**A**), 10 μM ABA (**B**) and 0.3 M mannitol (**C**). White arrows indicate the root length right before transferring. (**D**) Root length of 10-day-old Col-0 and organellar Tao3s-expressing seedlings grown on ½ MS medium. (**E**) Relative root length of Col-0 and organellar Tao3s-expressing seedlings from (**A**–**C**). Data was normalized to the relative root length at the control conditions (means ± SEM, *n* = 20). Significance was calculated using one-way ANOVA followed by Dunnett’s multiple comparisons test between Col-0 and organellar Tao3s-expressing seedlings (* *p* < 0.05; ** *p* < 0.01; ns, not significant).

## Data Availability

Not applicable.

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
