# Peer review of "Characterization of Organellar-Specific ABA Responses during Environmental Stresses in Tobacco Cells and Arabidopsis Plants"

_cells, 2022, doi:10.3390/cells11132039_

Round 1

Reviewer 1 Report

This is an important study on characterization of ABA in different organelles using ABA sensor targeted to ER, nucleus and the chloroplast as soluble probes. The study provides the evidence for real-time monitoring of ABA in organelles and plant response to environmental stresses. The manuscript is well written and understandable. Introduction aptly describes the importance of the study. Methods are clearly written. From Results onwards, authors have lost the focus of the study. Discussion section appears to be repetition of results, without giving proper importance of the ABA concentration in organelles and the responses to environmental stresses, which was the theme of the study. Authors should try to correlate the organ-specific ABA and plant response to environmental stress. This part should be emphasized and conclusion should be drawn accordingly. There is a need to expand the conclusion.

Additionally, minor changes as uniform formatting of all headings and sub-headings. Write tobacco as Nicotiana tabacum L.

Reviewer 2 Report

 - In this manuscript, the authors modified the previously developed ABA sensor ABAleon2.1_Tao3 (Tao3) and targeted it to different organelles including ER, chloroplast, and nucleus through addition of corresponding signal peptides.

 - The present research is well-conducted and provided useful findings. However, some minor revisions are required as shown below;

 - The abstract should focus more on the most important findings only.

 - The introduction should clearly discuss the hypothesis of the present study, including the recent literature. Many recent literature lacks here!

 - Material and Methods: The methods are well-planned and designed, however more details required on how the analyses have been done

 - Results are clear and well-represented. However, Figure 2 needs more explanation and discussion in relation to the other data.

 - In the discussion section, the data of the study should be discussed in relation to the literature findings and explain the reason of concordance and differences if present.

 -Please write a “conclusion section” and highlight the significant findings and future recommended studies.

 - The references section should be updated as per my above-mentioned comments.

Round 2

Reviewer 2 Report

The revised version is improved as per my suggested comments and revisions